# Trends and Associations of Past-30-Day Cigar Smoking in the U.S. by Age, Race/Ethnicity, and Sex, NSDUH 2002–2020

**DOI:** 10.3390/ijerph20186716

**Published:** 2023-09-06

**Authors:** Lauren R. Pacek, Michael D. Sawdey, Kimberly H. Nguyen, Maria Cooper, Eunice Park-Lee, Amy L. Gross, Elisabeth A. Donaldson, Karen A. Cullen

**Affiliations:** Office of Science, Center for Tobacco Products, Food and Drug Administration, Silver Spring, MD 20993, USA; michael.sawdey@fda.hhs.gov (M.D.S.); kimberly.nguyen@fda.hhs.gov (K.H.N.); maria.cooper1@fda.hhs.gov (M.C.); eunice.park-lee@fda.hhs.gov (E.P.-L.); amy.gross@fda.hhs.gov (A.L.G.); elisabeth.donaldson@fda.hhs.gov (E.A.D.); karen.cullen@fda.hhs.gov (K.A.C.)

**Keywords:** cigar, NSDUH, trends, race/ethnicity

## Abstract

Cigar smoking remains a public health issue in the United States (U.S.), with a heterogeneous prevalence based on sociodemographic characteristics. Nationally representative data suggest changes in cigar smoking over time, with some evidence for sociodemographic differences. Using data from the 2002–2019 National Survey on Drug Use and Health (NSDUH), the prevalence of past-30-day cigar smoking was examined overall and stratified by sociodemographic characteristics; joinpoint regression examined the trends. Logistic regression analyses identified the correlates of cigar smoking using 2020 NSDUH data. From 2002 to 2004, the prevalence of cigar smoking remained stable (5.33–5.73%), but declined from 2004 to 2019 (5.73–4.29%). Cigar smoking declined in some periods between 2002–2019 among the non-Hispanic White, Hispanic, ages 12–17, ages 18–20, ages 21–25, age ≥ 35, and male subgroups, but remained unchanged among the non-Hispanic Other, ages 26–34, and female subgroups. Cigar smoking increased among non-Hispanic Black persons overall from 2002 to 2019 (6.67–8.02%). Past-30-day cigarette smoking and drug or alcohol use disorder was associated with an increased likelihood of cigar use, while female sex was associated with a decreased likelihood of cigar use, across all age groups. Though a decline in the prevalence of past-30-day cigar smoking is seen in the general population, the same is not evident among all sociodemographic subgroups. Our findings have the potential to inform tobacco cessation efforts within clinical practice, as well as regulatory efforts to reduce cigar use.

## 1. Introduction

Cigar smoking is associated with significant morbidity—including oral and lung cancers, as well as heart disease [1,2,3]—and economic burden [4] in the United States (U.S.). It is also estimated that cigar smoking is responsible for approximately 9000 deaths annually in the U.S., representing 140,000 life-years lost [5]. Cigars remain a common form of combustible tobacco product use; data from the 2020 National Health Interview Survey (NHIS) indicated that cigar smoking was the third most common type of tobacco use among U.S. adults, with more than 8.6 million (3.5%) people aged ≥ 18 reporting smoking a cigar on some days or every day in the past month [6]. Only the use of cigarettes (12.5%) and electronic nicotine delivery system (ENDS) products (3.7%) surpassed cigar use among adults [6]. Similarly, data from the 2022 National Youth Tobacco Survey (NYTS) showed that cigars were the second most commonly used tobacco product among U.S. middle and high school students (1.9%), with 70,000 middle school students (0.6%) and 410,000 high school students (2.8%) reporting any past-30-day cigar smoking [7]. Furthermore, there is significant heterogeneity in cigar smoking when considering differences based on sociodemographic characteristics. 

In the 2020 NHIS data, 4.6% of non-Hispanic Black adults (aged ≥ 18) reported some-day or every-day cigar smoking, versus 3.8% of non-Hispanic White and 2.2% of Hispanic adults [6]. Similarly, the 2022 NYTS data indicate that 3.3% of non-Hispanic Black middle and high school students reported past-30-day cigar use, compared to 1.8% of non-Hispanic White and 1.7% of Hispanic students [7]. Furthermore, non-Hispanic Black youth and adults are more likely to report being an established cigar smoker (i.e., ever having used cigar products fairly regularly and currently using cigars some/every day) compared to other race/ethnicity groups [8,9]. Studies also indicate that the prevalence of cigar smoking among adults varies by other sociodemographic characteristics: young adults (ages 18–24) and males of any age are more likely to report current cigar smoking [10,11,12,13,14]. 

Recent nationally representative data have shown potential declines in overall cigar smoking prevalence [10,15], as well as for the use of certain types of cigars [16]. Data from the 2002–2019 National Survey on Drug Use and Health (NSDUH) Detailed Tables indicate that past-30-day cigar smoking has decreased significantly among youth aged 12–17 (4.5% to 1.4%), young adults aged 18–25 (11.0% to 7.7%), and adults aged ≥ 26 (4.6% to 4.0%) [15]. While these temporal trends were reported by age group, they were not further stratified by other sociodemographic characteristics of interest (e.g., race/ethnicity). Another analysis of NSDUH data (2010–2019) reported a decline in the prevalence of nonpremium cigar use (e.g., cigarillo and little cigar) among the general population, but an unchanged prevalence of premium cigar use over time [16]. These authors also analyzed trends in premium cigar use stratified by age group; slight increases in prevalence were observed among older groups (i.e., ages 50–64 and ≥65) and slight decreases in prevalence were observed among younger age groups [16]. Additional research utilizing data from the NSDUH identified significant increases in cigar use among non-Hispanic White males between 2002–2008; no other changes were observed based on race/ethnicity or sex [17]. However, a more recent analysis of the 2002–2016 NSDUH data [18] identified overall decreases in the prevalence of past-30-day cigar smoking among non-Hispanic Whites and Hispanic persons. No changes were observed among non-Hispanic Black and non-Hispanic Other race/ethnicity individuals [18]. It is worth noting that the cigar smoking prevalence estimates reported in this analysis were, on average, 1–2% higher than those in the NSDUH Detailed Tables, potentially due to additional exclusion criteria applied to the analytic sample [15,18].

The present study assessed temporal trends in cigar smoking in the U.S. general population overall, as well as stratified by sociodemographic characteristics; this study builds on prior work [18] by considering additional sociodemographic characteristics (e.g., sex, age, and race/ethnicity). We also build upon prior work [17,18] by extending the trend analyses (2002–2019) and utilizing additional analytic methods (i.e., joinpoint regression). The latter method enables us to analyze trends in cigar use, using joinpoint models where several different trend lines are connected at inflection points (or “joinpoints”) rather than relying on the assumption of a linear trend line. Further, the present study assessed the prevalence of cigar use in the 2020 NSDUH data. The data collected in 2020 could not be included in the trend analysis given the methodological changes. However, to describe U.S. cigar smokers using more recent data, this study assessed the use of other tobacco products and the sociodemographic characteristics of past-30-day cigar smokers in the 2020 NSDUH data.

## 2. Materials and Methods

### 2.1. Data Source and Study Population

Data came from the 2002–2020 NSDUH public use data files. The NSDUH is sponsored by the Substance Abuse Mental Health Services Administration (SAMHSA), and is a nationally representative, annual, cross-sectional survey assessing tobacco, alcohol, and drug use, mental health, and other health-related issues in the U.S. among those ages ≥ 12. Due to COVID-19-pandemic-related methodological changes in data collection in 2020 (i.e., web-based data collection was implemented in Quarter 4 of 2020), these data could not be included in trend analyses. Additional information regarding changes made to the NSDUH survey in 2020 can be found elsewhere [19]. Weighted interview response rates for the NSDUH survey years were as follows: 2002 (79%), 2003 (77%), 2004 (77%), 2005 (76%), 2006 (74%), 2007 (74%), 2008 (74%), 2009 (76%), 2010 (75%), 2011 (74%), 2012 (73%), 2013 (72%), 2014 (71%), 2015 (70%), 2016 (68%), 2017 (67%), 2018 (67%), 2019 (65%), and 2020 (60%). Detailed descriptions of sampling methods and survey techniques for the 2002–2019 [20] and 2020 [19] NSDUH are available elsewhere. This study was exempt from IRB review as NSDUH data are publicly available and all methods in this study were carried out in accordance with relevant guidelines and regulations.

### 2.2. Measures

Variable categorization and construction are described in Appendix A. Briefly, cigar smoking variables included past-30-day cigar smoking, age at first cigar use, and frequency of cigar smoking. The cigar items in the NSDUH questionnaire asked respondents about self-reported smoking of all or part of a cigar including “big cigars, cigarillos, and even little cigars that look like cigarettes”. Sociodemographic variables included sex, race/ethnicity, age, and family income. Current school grade was assessed only among individuals aged 12–17, while sexual identity and educational attainment were assessed among individuals aged ≥ 18. Other tobacco use variables included past-30-day cigarette smoking, past-30-day nicotine vaping, and past-30-day smokeless tobacco product use, defined as smoking a cigarette, vaping nicotine or tobacco with an e-cigarette or vaping device, and use of smokeless tobacco (i.e., snuff, dip, chewing tobacco, or snus), respectively, on ≥1 day during the past 30 days. People reporting past-30-day cigarette smoking were asked to indicate whether the cigarettes smoked during the past 30 days were menthol.

Other covariates included whether respondents had some form of health insurance, had any illicit drug/alcohol abuse/dependence within the past year, and general overall health quality. Individuals aged ≥ 18 reported lifetime service in the U.S. armed forces and completed the Kessler Psychological Distress Scale (K6) [21], a measure of nonspecific psychological distress (i.e., serious psychological distress; SPD) intended to be used in large, national health surveys to identify individuals with a high likelihood of having a diagnosable mental illness [22]. Scores were dichotomized, as in prior research [23].

### 2.3. Statistical Analysis

Analyses were conducted in SAS-callable SUDAAN (SAS version 9.4; SUDAAN version 11.0.4) unless otherwise noted. Analyses incorporated NSDUH sampling weights and controlled for the complex clustered sampling of the survey, which used the Taylor series estimation methods to provide accurate standard errors.

Weighted prevalence estimates of past-30-day cigar use and 95% confidence intervals were calculated for each year from 2002 to 2019 among the whole population, and stratified by select sociodemographic characteristics (i.e., age; race/ethnicity; sex; race/ethnicity and age; and sex and age). For analyses stratified by both race/ethnicity and age, as well as by both sex and age, we dichotomized the age variable (i.e., 12–20 versus ≥21) to avoid prohibitively small cell sizes that occured when including the age variable with 5 levels. 

Joinpoint (Joinpoint version 4.9.1.0 Surveillance Research Program, US National Cancer Institute) regression was utilized to characterize trends for all estimates. Joinpoint regression calculated trend joinpoints (i.e., inflection points) and annual percent change (APC) in each segment. The number of potential joinpoints is determined by the number of data points (i.e., years); for 18 years of data, the recommended maximum number of joinpoints is three [24]. We compared trends in past-30-day cigar smoking among two groups (e.g., between males and females) using pairwise comparisons by examining: (1) whether two trends were identical (i.e., test of coincidence); and (2) whether the two trends were parallel, allowing different intercepts (i.e., test of parallelism) [25,26]. A *p*-value of <0.05 for each test indicated that trends were significantly different from one another.

We also analyzed prevalence estimates and the number of cigar smokers for the 2020 data and conducted logistic regression modeling to assess whether sociodemographic characteristics were associated with past-30-day cigar smoking, stratified by age. Covariates in adjusted models were identified separately for youth (age 12–17) and adults (age ≥18) and were selected based on the literature and a priori theory, as well as statistical significance (*p* < 0.05) in bivariate analyses. Consideration was given to potential multicollinearity between covariates, investigated via examination of the correlation matrix, tolerance, and variance inflation; no evidence of multicollinearity was identified.

## 3. Results

### 3.1. Trends in Past 30-Day Cigar Smoking, Overall: 2002–2019

From 2002 to 2004, the prevalence of past-30-day cigar smoking among the overall U.S. population did not change (5.33% to 5.73%), but, from 2004 to 2019, the prevalence significantly decreased (5.73% to 4.29%) (Table 1).

### 3.2. Trends in Past 30-Day Cigar Smoking, Stratified by Age: 2002–2019

Among youth aged 12–17, past-30-day cigar smoking remained stable between 2002 and 2007 (4.42% to 4.33%) but decreased significantly between 2007 and 2019 (4.33% to 1.49%) (Table 1). Among young adults aged 18–20, past-30-day cigar smoking was stable from 2002 to 2004 (13.12% to 14.97%), declined significantly from 2004 to 2012 (14.97% to 12.15%), and declined more rapidly between 2012 and 2019 (12.15% to 7.00%). Among young adults aged 21–25, past-30-day cigar smoking was unchanged between 2002–2004 (9.57% to 11.55%) but decreased between 2004–2019 (11.55% to 8.11%). Conversely, cigar smoking remained unchanged between 2002 and 2019 among persons aged 26–34 (6.63% to 6.43%). Among adults aged ≥35, cigar smoking declined between 2002 and 2019 (3.95% to 3.55%). Pairwise parallel tests and pairwise coincidence tests indicated that the prevalence trends for each group were significantly different and not parallel or coincident (i.e., not identical) to one another (*p*’s < 0.05).

### 3.3. Trends in Past 30-Day Cigar Smoking, Stratified by Race/Ethnicity: 2002–2019

Among non-Hispanic White persons, past-30-day cigar smoking remained stable between 2002–2005 (5.41% to 5.85%) but declined between 2005–2019 (5.85% to 4.12%) (Table 2). Conversely, among non-Hispanic Black persons, cigar smoking increased between 2002–2019 (6.67% to 8.02%). Among Hispanic persons, cigar smoking decreased between 2002–2019 (4.97% to 3.09%). Past-30-day cigar smoking did not change over time among non-Hispanic Other persons (2.54% to 2.73%). Pairwise parallel tests and pairwise coincidence tests indicated that none of the prevalence trends for each race/ethnicity subgroup were parallel to one another (*p*’s < 0.05) nor identical to one another (*p* < 0.001).

### 3.4. Trends in Past 30-Day Cigar Smoking, Stratified by Sex: 2002–2019

Past-30-day cigar smoking remained unchanged between 2002–2005 among males (9.24% to 9.61%) but decreased between 2005–2019 (9.61% to 6.77%) (Table 3). Conversely, among females, cigar smoking remained relatively low and stable between 2002–2019 (1.67% to 1.96%). A parallel test and a coincidence test showed differences between prevalence trends in male and female respondents (*p’s* < 0.001), indicating that the two trends were neither parallel nor identical. 

### 3.5. Trends in Past 30-Day Cigar Smoking, Stratified by Race/Ethnicity and Age: 2002–2019

Among persons aged 12–20, past-30-day cigar smoking was stable between 2002–2008 (7.33% to 7.31%) and decreased between 2008–2019 (7.31% to 3.40%) (Table 3). Among respondents aged ≥21, cigar smoking decreased between 2002–2019 (4.95% to 4.44%). The pairwise parallel and the pairwise coincidence test indicated that prevalence trends for persons aged 12–20 and ≥21 were neither parallel nor identical (both *p’s* < 0.001).

Among non-Hispanic White persons aged 12–20, past-30-day cigar smoking was stable between 2002–2008 (8.38% to 8.57%) but decreased between 2008–2019 (8.57% to 3.76%) (Table 2). Among non-Hispanic White persons aged ≥21, cigar smoking decreased between 2002–2019 (4.92% to 4.17%). Among non-Hispanic Black persons aged 12–20, cigar smoking declined between 2002–2019 (6.37% to 4.78%). Conversely, among non-Hispanic Black persons aged ≥21, the prevalence of past-30-day cigar smoking increased between 2002–2019 (6.74% to 8.62%). Among Hispanic persons aged 12–20, cigar smoking remained stable between the years of 2002–2010 (5.55% to 5.36%) but declined between 2010–2019 (5.36% to 2.33%). Among Hispanic persons aged ≥ 21, the prevalence of cigar smoking declined between 2002–2019 (4.82% to 3.28%). Among non-Hispanic Other race/ethnicity persons aged 12–20, past-30-day cigar smoking declined between 2002–2019 (3.74% to 2.28%); the prevalence of cigar smoking remained unchanged among non-Hispanic Other persons aged ≥ 21 (2.29% to 2.81%). Pairwise parallel tests found that the trends were not parallel for the following groups: Hispanic persons aged 12–20 versus non-Hispanic Black persons aged 12–20 (*p* < 0.001), Hispanic persons aged ≥ 21 versus non-Hispanic Black persons aged ≥ 21 (*p* < 0.001), non-Hispanic Black persons aged 12–20 versus non-Hispanic White persons aged 12–20 (*p* = 0.008), and non-Hispanic Black persons aged ≥21 versus non-Hispanic White persons aged ≥ 21 (*p* < 0.001), and Hispanic persons aged ≥21 versus non-Hispanic Other participants aged ≥ 21 (*p* = 0.040). The remaining pairwise parallel tests were not significant. Pairwise coincidence tests show differences between prevalence trends between all race/ethnicity and age subgroups (*p*’s < 0.05), indicating that none of the trends were identical to one another.

### 3.6. Trends in Past 30-Day Cigar Smoking, Stratified by Sex and Age: 2002–2019

Among male respondents aged 12–20, the prevalence of cigar smoking remained stable between the years of 2002–2007 (10.59% to 11.51%) but declined between 2007–2019 (11.51% to 4.4%) (Appendix A). Among males aged ≥ 21, cigar smoking declined between 2002 and 2019 (8.97% to 7.16%). Among females aged 12–20, cigar smoking was unchanged between 2002–2011 (3.93% to 4.29%) but decreased from 2011–2019 (4.29% to 2.27%). Among female respondents aged ≥ 21, cigar smoking increased between 2002–2019 (1.28% to 1.91%). Pairwise parallel tests show differences between prevalence trends in male and female respondents, within age groups, indicating that the trends were not parallel to one another (age 12–20: *p* < 0.001; aged ≥ 21: *p* < 0.001). Pairwise coincidence tests show differences between age subgroups within sex, indicating that trends within sex were not identical to one another: age 12–20 versus ≥21 among females (*p* < 0.001); and age 12–20 versus ≥21 among males (*p* < 0.001).

### 3.7. Characteristics of Past 30-Day Cigar Smokers, Stratified by Age: 2020

In 2020, the prevalence of past-30-day cigar smoking was different between age groups (*p* < 0.001) and was most prevalent among adults aged 26–34 (6.26%), followed by adults aged 21–25 (6.08%), 18–20 (5.76%), and ≥35 (3.19%), and youth aged 12–17 (0.71%) (Appendix A). Among all age groups, respondents most commonly reported smoking cigars on 1–5 days during the past 30 days; there were no differences observed in the number of days on which respondents of all ages smoked cigars (*p* = 0.265). Cigarette smoking was equally prevalent between the age groups (*p* = 0.201), as was menthol cigarette smoking among those who reported smoking cigarettes during the past 30 days (*p* = 0.926). Past-30-day use of nicotine-containing ENDS products differed by age group (*p* < 0.001) and was most prevalent among youth aged 12–17 (39.11%), followed by adults aged 21–25 (32.02%), 18–20 (25.07%), 26–34 (8.15%), and ≥35 (4.82%). Past-30-day use of smokeless tobacco products was equally prevalent among all age groups (*p* = 0.535).

No differences were observed based on sex or race/ethnicity between age groups. Significant differences were found for family income (*p* < 0.001) between age groups. Overall health status differed between age groups (*p* < 0.001) where adults aged ≥35 were most likely to report fair/poor health (24.26%), followed by adults aged 26–34 (12.96%), 21–25 (9.27%), and 18–20 (4.53%), and youth aged 12–17 (5.26%). Having health insurance was least prevalent among adults aged 26–34 (68.18%) but was more prevalent among all other age groups (12–17: 92.88%; 18–20: 90.72%; 21–25: 84.28%; ≥35: 86.33; *p* = 0.019). Past-year drug or alcohol use disorders also differed by age (*p* < 0.001) and were most prevalent among youth aged 12–17 (61.14%), followed by adults aged 21–25 (43.49%), 18–20 (30.29%), 26–34 (21.50%), and ≥35 (20.53).

Among adult age groups, differences in age groups were observed based on sexual identity (*p* < 0.001), but not completed education (*p* = 0.270). Lifetime military service differed significantly by age group (*p* < 0.001) and was significantly more prevalent among adults aged ≥35 (20.70%), followed by 26–34 (3.41%), and 21–25 (2.34%); no respondents aged 18–20 reported any lifetime military service. Last, the prevalence of past-month SPD differed by age (*p* = 0.006); SPD was most prevalent among persons aged 21–25 (26.84%), followed by 18–20 (17.70%), ≥35 (12.27%), and 26–34 (11.35%).

### 3.8. Correlates of Past 30-Day Cigar Smoking, Stratified by Age: 2020

Ages 12–17. In adjusted models, youth who reported past-30-day cigarette smoking were 9.01 times (95% CI: 2.85–28.52) more likely to also report past-30-day cigar smoking than those who did not report smoking cigarettes (Table 4). Persons reporting Hispanic ethnicity were less likely than non-Hispanic White race/ethnicity persons to report past-30-day cigar smoking (aOR: 0.17; 95% CI: 0.04–0.81), as were female respondents, as compared to males (aOR: 0.43; 95% CI: 0.19–0.98). Youths in grades 9–12 were more likely to report past-30-day cigar smoking as compared to those in grade 8 or below (aOR: 6.77; 95% CI: 2.44–18.81). Additionally, youths reporting a past-year drug or alcohol use disorder were more likely than those not reporting such comorbidity to report past-30-day cigar use (aOR: 26.92; 95% CI: 8.98–80.66).

Ages 18–20. Individuals reporting past-30-day cigarette smoking were 2.31 times (95% CI: 1.02–5.52) more likely to also report past-30-day cigar use, as compared to those not reporting cigarette smoking. Persons reporting Hispanic ethnicity were less likely than non-Hispanic White persons to report past-30-day cigar smoking (aOR: 0.20, 95% CI: 0.07–0.54). Female respondents were less likely than males to report past-30-day cigar smoking (aOR: 0.45, 95% CI: 0.20–1.00), while persons reporting a past-year drug or alcohol use disorder were more likely than those without such a disorder to report cigar smoking (aOR: 3.26, 95% CI: 1.19–8.92).

Ages 21–25. Persons reporting cigarette smoking (aOR: 5.26, 95% CI: 3.43–8.06) and nicotine ENDS product use (aOR: 2.46, 95% CI: 1.33–4.57) were more likely to report past-30-day cigar smoking than were persons who did not report cigarette or nicotine ENDS use. Non-Hispanic Black persons, as compared to non-Hispanic White, were three times more likely to report past-30-day cigar smoking (aOR: 3.22, 95% CI: 1.97–5.27). Female persons were less likely than male individuals to report past-30-day cigar smoking (aOR: 0.58, 95% CI: 0.39–0.87). Persons reporting past-year SPD (aOR: 1.91, 95% CI: 1.20–3.02) and past-year drug/alcohol use disorder (aOR: 2.58, 95% CI: 1.67–4.01) were more likely to report past-30-day cigar smoking, as compared to people without SPD or drug/alcohol use disorder, respectively.

Ages 26–34. People who reported past-30-day cigarette smoking were also more likely to report past-30-day cigar smoking, as compared to those who did not smoke cigarettes (aOR: 3.41, 95% CI: 2.22–5.22). Non-Hispanic Black persons, as compared to non-Hispanic White, were more than two times as likely to report past-30-day cigar smoking (aOR: 2.63, 95% CI: 1.45–4.75). Female persons, as compared to male, were less likely to report past-30-day cigar smoking (aOR: 0.41, 95% CI: 0.25–0.67), while people with past-year drug or alcohol use disorder were more likely to report past-30-day cigar smoking (aOR: 1.70, 95% CI: 1.13–2.58).

Ages ≥35. People reporting past-30-day cigarette smoking (aOR: 4.17, 95% CI: 2.80–6.19), as compared to people who did not report smoking, and non-Hispanic Black persons (aOR: 2.27, 95% CI: 1.47–3.52), as compared to non-Hispanic White, were more likely to report past-30-day cigar smoking. Female persons (aOR: 0.27, 95% CI: 0.19–0.40), as compared to male, and persons reporting being gay or lesbian (aOR: 0.18, 95% CI: 0.05–0.76), as compared to heterosexual, were less likely to report past-30-day cigar smoking. Persons who reported past-month SPD (aOR: 1.88, 95% CI: 1.11–3.20), as compared to no SPD, and past-year drug or alcohol use disorder (aOR: 2.63, 95% CI: 1.56–4.44), as compared to no such disorder, were more likely to report past-30-day cigar smoking.

## 4. Discussion

Over the past two decades, the prevalence of past-30-day cigar smoking has declined significantly among the U.S. general population. Specifically, though the prevalence of cigar smoking remained unchanged between 2002 and 2004, it decreased significantly between 2004 and 2019. This general pattern is also evident among numerous sociodemographic subgroups within the sample. Though this is indicative of a public health victory, declines in the prevalence of cigar smoking are not mirrored among all population subgroups. Additionally, though the prevalence of cigar smoking decreased over time in terms of percentage of the overall population, approximately 11.8 million people aged 12 years or older reported past-30-day cigar smoking in 2019. When considering increases in the overall size of the U.S. population during the study period (2002 to 2019), this likely represents only a small decrease in cigar use since 2002, when approximately 12.5 million people reported past-30-day cigar use. 

Findings from the present study highlight significant increases in the prevalence of past-30-day cigar smoking among individuals identifying as non-Hispanic Black. While racial/ethnic differences in the cross-sectional prevalence of cigar use are well-documented—with cigar-smoking being significantly more prevalent among individuals of non-Hispanic Black race/ethnicity [6,10]—an increasing prevalence of cigar smoking in this group over time is troubling. When analyses were further stratified by age, a significant increase in past-30-day cigar use was evident among non-Hispanic Black persons aged ≥ 21, specifically. Moreover, a significant decline in the prevalence of cigar use was observed among non-Hispanic Black individuals aged 12–20, similar to the trend seen in the overall population. However, it is worth noting that the declines among this race/age group were relatively slower than those observed among other race/ethnicity group individuals aged 12–20. These findings are consistent with non-Hispanic Black individuals bearing a disproportionately high burden of morbidity and mortality from tobacco-related diseases relative to non-Hispanic White persons. Additionally, the present analysis identified an increase in past-30-day cigar use among female persons aged ≥ 21. Though the overall prevalence of cigar use remained quite low in this age group and increased less than 1% between 2002 and 2019, it should be noted that this represents an increase in approximately one million cigar users during this time period. Regarding increases in cigar use prevalence among non-Hispanic Black individuals, studies have found that cigars are disproportionately marketed toward Black individuals [27,28], and that cigars are marketed in retail outlets in neighborhoods with large proportions of Black residents [29,30,31]. Overall, these findings suggest a possible benefit for tailored interventions among those identifying as non-Hispanic Black and female persons aged ≥ 21. Future research can provide additional insight into the social, cultural, and/or economic factors that may explain these documented disparities regarding non-Hispanic Black and female respondents aged ≥ 21. Continued surveillance efforts among these and other groups can contribute to tobacco control strategies and, with Food and Drug Administration (FDA) regulation, may help reduce tobacco use. In fact, in April 2022, FDA announced a proposed product standard that would prohibit characterizing flavors (other than tobacco) in cigars [32]. 

Prior studies have also reported changes in the prevalence of past-30-day cigar smoking over time among various subgroups of the population [15,17,18]. Most recently, utilizing data from the 2002–2016 NSDUH, Weinberger and colleagues [18] described significant declines in the prevalence of past-30-day cigar smoking among Hispanic and non-Hispanic White individuals, but no changes in prevalence among persons reporting non-Hispanic Black or non-Hispanic Other race/ethnicity. Differences in analytic methodologies between the present analysis—which utilized the same variable definitions and denominators as the NSDUH Detailed Tables [15]—and Weinberger et al. [18] possibly account for these discrepant findings.

Although direct comparisons cannot be made between the data from 2020 and prior years, it is worth noting that approximately 10.4 million people were cigar smokers in 2020, representing a sizeable portion of the U.S. population. When examining the prevalence and correlates of past-30-day cigar use among the various age groups, several notable findings emerged. Past-30-day cigar use in 2020 was most prevalent among the 21–25 and 26–34 age groups, followed closely by those aged 18–20. These findings are consistent with prior work, which has found cigar use to be most prevalent among young adult age groups [10,11,12,13]. Consistent with prior research indicating the common nature of dual and multiple tobacco product use [33], persons in all age groups who reported past-30-day cigarette smoking were significantly more likely to also be past-30-day cigar smokers. Prior research has consistently identified cigar use as being more common among individuals reporting non-Hispanic Black race/ethnicity [6,8,10]. Congruent findings were borne out in the present analyses, with non-Hispanic Black respondents being more likely than non-Hispanic White persons to report past-30-day cigar smoking, though these findings were statistically significant only among persons aged 21–25, 26–34, and 35 and older. It is likely that the lack of statistical significance among younger age groups is due to small sample sizes (ages 12–17, n = 57; ages 18–20, n = 123). Among the two youngest age groups—ages 12–17 and 18–20—persons reporting Hispanic ethnicity were approximately 80% less likely than non-Hispanic White persons to report current cigar use. Associations between Hispanic ethnicity and cigar use were not observed among the older age groups, though the adjusted odds ratios for these groups—while not significant—were less than 1, indicating a potential similar negative association among the older age groups. This finding is consistent with those from an analysis of Wave 3 adult data of the Population Assessment of Tobacco and Health (PATH) study, which also found no association between Hispanic ethnicity—as compared to non-Hispanic White race/ethnicity—and past-30-day cigar smoking [8]. These findings have the potential to inform targeted public health interventions. Ultimately, given that the 2020 survey occurred during the height of the COVID-19 pandemic—and utilized different data collection methods—these results should perhaps be interpreted with some caution. Prior research indicates that the experience of living through a pandemic—whether through increased general stress or otherwise—may have impacted the prevalence and frequency of tobacco use [34,35].

Several limitations should be acknowledged. Data were collected via self-report, which is subject to social desirability and recall biases. NSDUH also does not distinguish between individual types of cigars (e.g., cigarillos or filtered cigars), instead asking about cigars as a homogenous product category. This limits our ability to assess the potential differences in cigar type trends and associated correlates, as some recent research suggests that differences exist based on cigar type [16,36]. Ultimately, it would be beneficial to conduct large-scale, nationally representative surveys to collect information regarding the specific type(s) of cigar(s) used by respondents, rather than assessing them with a single “cigar” item. Dichotomizing the age variable in analyses with race-by-age and sex-by-age interaction terms, while done to prevent small cell sizes, may also result in a potential loss of information and missing non-linear relationships. Given that NSDUH is an annual, cross-sectional survey, we are unable to assess the trajectories of cigar use among individuals, though this was outside of the scope of the present paper. Additionally, due to differences in the data collection methodology that occurred due to the COVID-19 pandemic, we were unable to include the 2020 data in the trend analyses. Despite the limitations, this study represents the most current, comprehensive assessment of trends in past-30-day cigar smoking prevalence among the U.S. general population.

## 5. Conclusions

While the prevalence of cigar use declined among the U.S. general population during 2002–2019, this study identified specific subpopulations in which cigar use remained stable or increased. In fact, past-30-day cigar use increased significantly among non-Hispanic Black individuals—particularly among those aged ≥21—and among female respondents aged ≥ 21. We also identified correlates of past-30-day cigar use, including cigarette smoking, non-Hispanic Black race/ethnicity, male sex, past-month SPD, and drug or alcohol use disorder, which were largely consistent across age groups. Given the continually evolving nature of the tobacco product landscape, information on trends and correlates of cigar smoking can inform efforts to reduce cigar use.

## Figures and Tables

**Table 1 ijerph-20-06716-t001:** Number, percentage, and trend of current (past-30-day) cigar smoking, by age—National Survey on Drug Use and Health, United States, 2002–2019.

**Estimated Number of Cigar Smokers**																								
	**2002**	**2003**	**2004**	**2005**	**2006**	**2007**	**2008**	**2009**	**2010**	**2011**	**2012**	**2013**	**2014**	**2015**	**2016**	**2017**	**2018**	**2019**	**Joinpoint Regression Trends ^a^**					
	**Weighted N**	**Weighted N**	**Weighted N**	**Weighted N**	**Weighted N**	**Weighted N**	**Weighted N**	**Weighted N**	**Weighted N**	**Weighted N**	**Weighted N**	**Weighted N**	**Weighted N**	**Weighted N**	**Weighted N**	**Weighted N**	**Weighted N**	**Weighted N**	**Year**	**APC ^b^** **(95% CI) ^c^**	**Year**	**APC** **(95% CI)**	**Year**	**APC** **(95% CI)**
Total	12,535,430	12,744,160	13,783,780	13,623,550	13,751,740	13,264,710	13,504,580	13,255,340	13,376,990	12,773,820	13,809,180	12,441,360	12,062,450	12,392,840	12,250,060	12,528,720	12,200,270	11,818,470	-	-	-	-	-	-
Age 12–17	1,093,440	1,120,960	1,159,650	1,072,320	1,067,340	1,093,940	942,230	951,260	779,510	876,130	681,860	558,660	526,060	488,310	437,770	507,740	418,920	371,540	-	-	-	-	-	-
Age 18–20	1,636,170	1,680,000	1,867,440	1,903,420	1,760,220	1,924,460	1,845,320	1,763,410	1,741,710	1,705,060	1,599,310	1,520,540	1,372,250	1,234,480	1,183,180	1,178,400	1,109,410	923,340	-	-	-	-	-	-
Age 21–25	1,776,310	1,986,190	2,277,720	2,049,060	2,233,970	2,008,580	1,997,100	2,106,620	2,154,300	2,051,310	2,165,520	1,976,430	2,115,720	1,810,930	1,803,270	1,929,130	1,783,460	1,666,390	-	-	-	-	-	-
Age 26–34	2,330,980	2,323,810	2,281,020	2,356,360	2,907,210	2,414,260	2,527,300	2,701,610	2,497,950	2,670,510	2,713,200	2,901,100	2,597,180	2,965,360	2,655,410	2,842,820	2,746,950	2,592,740	-	-	-	-	-	-
≥Age 35	5,698,530	5,633,200	6,197,960	6,242,390	5,782,300	5,823,470	6,192,630	5,732,450	6,203,520	5,470,820	6,649,290	5,484,620	5,451,240	5,893,770	6,170,430	6,070,340	6,141,530	6,264,460	-	-	-	-	-	-
**Prevalence of Cigar Smoking**																								
	**2002**	**2003**	**2004**	**2005**	**2006**	**2007**	**2008**	**2009**	**2010**	**2011**	**2012**	**2013**	**2014**	**2015**	**2016**	**2017**	**2018**	**2019**	**Joinpoint regression trends**					
	**% (95% Cl)**	**% (95% Cl)**	**% (95% Cl)**	**% (95% Cl)**	**% (95% Cl)**	**% (95% Cl)**	**% (95% Cl)**	**% (95% Cl)**	**% (95% Cl)**	**% (95% Cl)**	**% (95% Cl)**	**% (95% Cl)**	**% (95% Cl)**	**% (95% Cl)**	**% (95% Cl)**	**% (95% Cl)**	**% (95% Cl)**	**% (95% Cl)**	**Year**	**APC** **(95% CI)**	**Year**	**APC** **(95% CI)**	**Year**	**APC** **(95% CI)**
Total	5.33 (5.02–5.66)	5.36 (5.07–5.67)	5.73 (5.40–6.09)	5.60 (5.30–5.92)	5.59 (5.29–5.90)	5.35 (5.06–5.66)	5.41 (5.05–5.78)	5.26 (4.99–5.56)	5.27 (4.98–5.58)	4.96 (4.70–5.24)	5.31 (4.94–5.70)	4.74 (4.41–5.09)	4.55 (4.33–4.79)	4.63 (4.40–4.87)	4.55 (4.33–4.78)	4.60 (4.31–4.91)	4.46 (4.24–4.69)	4.29 (4.06–4.54)	2002–2004	4.5 (−3.6, 13.3)	**2004–2019**	**−1.9** **(−2.2, −1.6)**	-	-
Age 12–17	4.42 (4.11–4.75)	4.48 (4.11–4.90)	4.60 (4.20–5.03)	4.23 (3.84–4.66)	4.20 (3.87–4.57)	4.33 (4.00–4.70)	3.79 (3.37–4.25)	3.87 (3.49–4.28)	3.20 (2.89–3.54)	3.51 (3.21–3.84)	2.73 (2.45–3.05)	2.24 (1.97–2.55)	2.11 (1.85–2.42)	1.96 (1.68–2.29)	1.76 (1.52–2.03)	2.04 (1.75–2.37)	1.68 (1.45–1.95)	1.49 (1.24–1.80)	2002–2007	−0.6 (−4.5, 3.5)	**2007–2019**	**−8.6** **(−10.1, −6.9)**	-	-
Age 18–20	13.12 (12.14–14.16)	13.46 (12.59–14.38)	14.97 (13.81–16.21)	14.64 (13.64–15.69)	13.82 (12.98–14.81)	14.54 (13.28–15.91)	13.92 (12.97–14.92)	13.06 (12.05–14.14)	12.91 (11.98–13.91)	12.69 (11.68–13.77)	12.15 (11.23–13.13)	11.50 (10.56–12.51)	10.47 (9.44–11.59)	9.48 (8.36–10.73)	9.36 (8.54–10.26)	9.18 (8.20–10.26)	8.54 (7.45–9.76)	7.00 (6.25–7.83)	2002–2004	7.6 (−1.8, 17.9)	**2004–2012**	**−2.3** **(−3.6, −1.1)**	**2012–2019**	**−6.9** **(−8.5, −5.2)**
Age 21–25	9.57 (8.84–10.36)	10.32 (9.52–11.18)	11.55 (10.64–12.52)	10.52 (9.71–11.38)	11.17 (10.43–11.95)	10.30 (9.66–10.98)	10.15 (9.28–11.08)	10.49 (9.75–11.28)	10.47 (9.47–11.55)	9.83 (9.09–10.63)	10.11 (9.41–10.86)	9.17 (8.39–10.00)	9.69 (8.89–10.56)	8.27 (7.59–9.01)	8.22 (7.50–9.01)	8.99 (8.15–9.90)	8.48 (7.74–9.28)	8.11 (7.34–8.96)	2002–2004	9.1 (−4.9, 25.1)	**2004–2019**	**−2.1** **(−2.7, −1.5)**	-	-
Age 26–34	6.63 (5.76–7.62)	6.65 (5.78–7.63)	6.52 (5.72–7.42)	6.76 (5.95–7.67)	8.28 (7.20–9.51)	6.84 (6.01–7.77)	7.09 (6.23–8.06)	7.46 (6.45–8.61)	6.84 (6.13–7.63)	7.33 (6.43–8.35)	7.35 (6.42–8.40)	7.77 (6.73–8.95)	6.86 (6.22–7.55)	7.74 (6.96–8.59)	6.84 (6.21–7.53)	7.19 (6.40–8.06)	6.86 (6.19–7.61)	6.43 (5.80–7.12)	2002–2019	0.0 (−0.6, 0.7)	-	-	-	-
≥Age 35	3.95 (3.57–4.38)	3.86 (3.50–4.25)	4.18 (3.73–4.69)	4.15 (3.72–4.62)	3.79 (3.39–4.23)	3.77 (3.41–4.16)	3.96 (3.51–4.47)	3.64 (3.31–4.01)	3.91 (3.48–4.39)	3.38 (3.02–3.78)	4.06 (3.60–4.58)	3.32 (2.94–3.74)	3.26 (2.97–3.57)	3.48 (3.17–3.81)	3.61 (3.26–3.98)	3.50 (3.20–3.84)	3.51 (3.22–3.84)	3.55 (3.24–3.89)	**2002–2019**	**−0.9** **(−1.4, −0.4)**	-	-	-	-

^a^ Joinpoint regression is used to identify statistically significant trend change points and the APC in each trend segment using a Monte Carlo permutation method allowing for a maximum of 3 joinpoints. ^b^ APC = Annual Percentage Change; bold text indicates significance at the *p* < 0.05 level. ^c^ 95% CI = 95% confidence interval.

**Table 2 ijerph-20-06716-t002:** Number, percentage, and trend of current (past-30-day) cigar smoking, by race/ethnicity group and age—National Survey on Drug Use and Health, United States, 2002–2019.

**Estimated Number of Cigar Smokers**																								
	**2002**	**2003**	**2004**	**2005**	**2006**	**2007**	**2008**	**2009**	**2010**	**2011**	**2012**	**2013**	**2014**	**2015**	**2016**	**2017**	**2018**	**2019**	**Joinpoint regression trends ^a^**					
	**Weighted N**	**Weighted N**	**Weighted N**	**Weighted N**	**Weighted N**	**Weighted N**	**Weighted N**	**Weighted N**	**Weighted N**	**Weighted N**	**Weighted N**	**Weighted N**	**Weighted N**	**Weighted N**	**Weighted N**	**Weighted N**	**Weighted N**	**Weighted N**	**Year**	**APC ^b^** **(95% CI) ^c^**	**Year**	**APC** **(95% CI)**	**Year**	**APC** **(95% CI)**
Total	12,535,430	12,744,160	13,783,780	13,623,550	13,751,740	13,264,710	13,504,580	13,255,340	13,376,990	12,773,820	13,809,180	12,441,360	12,062,450	12,392,840	12,250,060	12,528,720	12,200,270	11,818,470	-	-	-	-	-	-
Age 12–20	2,729,610	2,800,960	3,027,090	2,975,750	2,827,560	3,018,400	2,787,550	2,714,670	2,521,220	2,581,190	2,281,180	2,079,210	1,898,300	1,722,780	1,620,950	1,686,140	1,528,330	1,294,880	-	-	-	-	-	-
≥Age 21	9,805,820	9,943,200	10,756,690	10,647,810	10,924,180	10,246,310	10,717,030	10,540,670	10,855,770	10,192,630	11,528,010	10,362,160	10,164,140	10,670,060	10,629,110	10,842,580	10,671,940	10,523,590	-	-	-	-	-	-
NH ^d^ White	8,949,600	8,832,140	10,016,300	9,809,100	9,582,530	9,443,120	9,388,340	8,902,130	8,953,340	8,446,630	9,563,200	8,160,240	7,954,450	7,454,280	7,826,190	7,890,910	7,358,000	7,033,340	-	-	-	-	-	-
Age 12–20	1,967,430	1,949,410	2,082,020	2,076,270	2,011,800	2,249,380	1,947,980	1,915,040	1,694,080	1,763,560	1,551,300	1,405,000	1,276,140	980,300	1,056,570	1,123,240	970,550	741,820	-	-	-	-	-	-
≥Age 21	6,982,170	6,882,730	7,934,280	7,732,830	7,570,730	7,193,740	7,440,360	6,987,090	7,259,260	6,683,070	8,011,910	6,755,240	6,678,310	6,473,980	6,769,620	6,767,660	6,387,460	6,291,420	-	-	-	-	-	-
NH Black	1,788,080	1,983,800	1,685,570	1,893,130	2,114,300	1,898,400	2,090,540	2,088,970	2,263,590	2,245,140	2,069,830	2,102,150	2,066,490	2,513,370	2,451,000	2,437,750	2,735,620	2,665,820	-	-	-	-	-	-
Age 12–20	337,350	362,840	375,350	379,740	346,790	298,880	322,030	291,390	321,300	307,580	253,380	231,080	236,700	285,270	257,690	248,920	228,660	247,940	-	-	-	-	-	-
≥Age 21	1,450,730	1,620,960	1,310,220	1,513,380	1,767,510	1,599,520	1,768,510	1,797,580	1,942,290	1,937,550	1,816,450	1,871,070	1,829,780	2,228,100	2,193,310	2,188,830	2,506,960	2,417,880	-	-	-	-	-	-
Hispanic	1,445,140	1,466,110	1,524,330	1,393,380	1,630,850	1,391,870	1,617,300	1,739,780	1,655,200	1,455,260	1,543,460	1,553,960	1,540,040	1,644,560	1,350,540	1,597,100	1,484,500	1,467,840	-	-	-	-	-	-
Age 12–20	332,740	353,520	452,930	408,420	347,360	340,520	411,780	384,410	409,790	396,450	342,920	310,760	283,770	342,980	215,410	208,510	234,520	218,770	-	-	-	-	-	-
≥Age 21	1,112,400	1,112,590	1,071,400	984,960	1,283,490	1,051,350	1,205,520	1,355,360	1,245,410	1,058,810	1,200,550	1,243,200	1,256,270	1,301,590	1,135,120	1,388,590	1,249.980	1,249,070	-	-	-	-	-	-
NH Other	352,620	462,120	557,590	527,950	424,060	531,320	408,410	524,460	504,860	626,790	632,690	625,010	501,480	780,630	622,340	602,970	622,140	651,570	-	-	-	-	-	-
Age 12–20	92,090	135,200	116,790	111,310	121,620	129,620	105,770	123,830	96,060	113,590	133,580	132,370	101,690	114,240	91,280	105,460	94,610	86,360	-	-	-	-	-	-
≥Age 21	260,530	326,920	440,800	416,630	302,450	401,700	302,650	400,630	408,800	513,200	499,110	492,640	399,780	666,390	531,060	497,500	527,540	565,210	-	-	-	-	-	-
**Prevalence of Cigar Smoking**																								
	**2002**	**2003**	**2004**	**2005**	**2006**	**2007**	**2008**	**2009**	**2010**	**2011**	**2012**	**2013**	**2014**	**2015**	**2016**	**2017**	**2018**	**2019**	**Joinpoint regression trends**					
	**% (95% Cl)**	**% (95% Cl)**	**% (95% Cl)**	**% (95% Cl)**	**% (95% Cl)**	**% (95% Cl)**	**% (95% Cl)**	**% (95% Cl)**	**% (95% Cl)**	**% (95% Cl)**	**% (95% Cl)**	**% (95% Cl)**	**% (95% Cl)**	**% (95% Cl)**	**% (95% Cl)**	**% (95% Cl)**	**% (95% Cl)**	**% (95% Cl)**	**Year**	**APC** **(95% CI)**	**Year**	**APC** **(95% CI)**	**Year**	**APC** **(95% CI)**
Total	5.33 (5.02–5.66)	5.36 (5.07–5.67)	5.73 (5.40–6.09)	5.60 (5.30–5.92)	5.59 (5.29–5.90)	5.35 (5.06–5.66)	5.41 (5.05–5.78)	5.26 (4.99–5.56)	5.27 (4.98–5.58)	4.96 (4.70–5.24)	5.31 (4.94–5.70)	4.74 (4.41–5.09)	4.55 (4.33–4.79)	4.63 (4.40–4.87)	4.55 (4.33–4.78)	4.60 (4.31–4.91)	4.46 (4.24–4.69)	4.29 (4.06–4.54)	2002–2004	4.5 (−3.6, 13.3)	**2004–2019**	**−1.9** **(−2.2, −1.6)**	-	-
Age 12–20	7.33 (6.92–7.77)	7.47 (7.10–7.87)	8.03 (7.47–8.63)	7.76 (7.36–8.18)	7.42 (7.02–7.83)	7.85 (7.34–8.38)	7.31 (6.83–7.81)	7.12 (6.64–7.64)	6.66 (6.26–7.09)	6.72 (6.27–7.19)	5.99 (5.56–6.45)	5.46 (5.07–5.87)	5.00 (4.54–5.50)	4.54 (4.10–5.03)	4.32 (3.98–4.69)	4.46 (4.07–4.89)	4.03 (3.61–4.50)	3.40 (3.12–3.70)	2002–2008	0.5 (−1.2, 2.2)	**2008–2019**	**−6.7** **(−7.5, −5.8**)	-	-
≥Age 21	4.95 (4.59–5.35)	4.97 (4.65–5.31)	5.30 (4.93–5.70)	5.20 (4.85–5.56)	5.25 (4.93–5.60)	4.89 (4.57–5.24)	5.06 (4.67–5.49)	4.93 (4.62–5.26)	5.03 (4.70–5.38)	4.65 (4.35–4.98)	5.19 (4.78–5.64)	4.62 (4.25–5.02)	4.47 (4.23–4.74)	4.64 (4.40–4.91)	4.58 (4.32–4.86)	4.63 (4.32–4.96)	4.52 (4.28–4.78)	4.44 (4.18–4.72)	**2002–2019**	**−0.9** **(−1.2, −0.6)**	-	-	-	-
NH White	5.41 (5.02–5.83)	5.31 (5.01–5.63)	6.00 (5.59–6.42)	5.85 (5.50–6.22)	5.69 (5.35–6.06)	5.59 (5.23–5.96)	5.54 (5.13–5.98)	5.24 (4.90–5.60)	5.27 (4.92–5.63)	4.99 (4.59–5.44)	5.64 (5.13–6.20)	4.80 (4.35–5.30)	4.67 (4.43–4.93)	4.37 (4.04–4.72)	4.58 (4.29–4.89)	4.62 (4.28–4.99)	4.31 (4.00–4.64)	4.12 (3.83–4.43)	2002–2005	3.5 (−2.5, 10.0)	**2005–2019**	**−2.4** **(−2.9, −1.8)**	-	-
Age 12–20	8.38 (7.82–8.98)	8.37 (7.82–8.95)	8.95 (8.23–9.73)	8.83 (8.33–9.36)	8.73 (8.20–9.31)	9.67 (9.01–10.37)	8.57 (7.94–9.25)	8.51 (7.81–9.28)	7.71 (7.17–8.28)	8.27 (7.71–8.86)	7.39 (6.67–8.18)	6.77 (6.22–7.36)	6.22 (5.50–7.04)	4.83 (4.23–5.52)	5.29 (4.66–5.99)	5.65 (5.01–6.37)	4.93 (4.35–5.59)	3.76 (3.29–4.30)	2002–2008	1.7 (−1.0, 4.5)	**2008–2019**	**−6.6** **(−8.1, −5.1)**	-	-
≥Age 21	4.92 (4.48–5.40)	4.81 (4.48–5.17)	5.52 (5.07–6.00)	5.36 (4.96–5.78)	5.21 (4.84–5.60)	4.93 (4.54–5.37)	5.07 (4.61–5.58)	4.74 (4.38–5.13)	4.90 (4.53–5.30)	4.52 (4.06–5.04)	5.39 (4.85–5.99)	4.53 (4.03–5.08)	4.46 (4.19–4.74)	4.31 (3.94–4.71)	4.49 (4.17–4.84)	4.49 (4.13–4.87)	4.23 (3.90–4.59)	4.17 (3.85–4.51)	**2002–2019**	**−1.3** **(−1.7, −0.8)**	-	-	-	-
NH Black	6.67 (5.63–7.87)	7.23 (6.30–8.28)	6.02 (5.22–6.93)	6.62 (5.74–7.62)	7.26 (6.36–8.28)	6.49 (5.66–7.44)	7.07 (6.11–8.17)	6.95 (6.03–8.00)	7.49 (6.36–8.80)	7.42 (6.44–8.52)	6.74 (5.72–7.93)	6.73 (5.70–7.93)	6.52 (5.72–7.43)	7.84 (7.09–8.65)	7.60 (6.86–8.42)	7.42 (6.75–8.14)	8.30 (7.39–9.32)	8.02 (7.18–8.95)	**2002–2019**	**1.1** **(0.5, 1.7)**	-	-	-	-
Age 12–20	6.37 (5.60–7.25)	6.55 (5.60–7.66)	6.73 (5.59–8.09)	6.71 (5.79–7.77)	6.06 (5.09–7.19)	5.27 (4.12–6.72)	5.54 (4.44–6.90)	5.10 (4.13–6.28)	5.75 (4.84–6.82)	5.67 (4.73–6.80)	4.66 (3.82–5.68)	4.30 (3.37–5.46)	4.52 (3.44–5.91)	5.30 (4.23–6.63)	4.94 (4.02–6.05)	4.69 (3.73–5.88)	4.37 (3.34–5.69)	4.78 (3.86–5.89)	**2002–2019**	**−2.3** **(−3.0, −1.6**)	-	-	-	-
≥Age 21	6.74 (5.48–8.27)	7.40 (6.25–8.73)	5.84 (4.89–6.97)	6.60 (5.56–7.80)	7.56 (6.48–8.80)	6.79 (5.82–7.90)	7.45 (6.33–8.75)	7.38 (6.24–8.71)	7.88 (6.57–9.43)	7.80 (6.61–9.17)	7.19 (5.98–8.62)	7.24 (6.01–8.69)	6.92 (6.00–7.97)	8.35 (7.50–9.28)	8.12 (7.26–9.07)	7.94 (7.13–8.84)	9.05 (7.99–10.23)	8.62 (7.64–9.71)	**2002–2019**	**1.5** **(0.9, 2.2)**	-	-	-	-
Hispanic	4.97 (4.17–5.91)	4.91 (4.02–5.97)	4.91 (4.09–5.88)	4.34 (3.71–5.07)	4.88 (4.02–5.91)	4.06 (3.29–5.01)	4.61 (3.74–5.67)	4.84 (4.12–5.68)	4.50 (3.76–5.38)	3.69 (3.11–4.38)	3.83 (3.11–4.71)	3.77 (3.17–4.47)	3.63 (3.13–4.19)	3.78 (3.23–4.40)	3.05 (2.59–3.58)	3.49 (2.84–4.29)	3.18 (2.76–3.67)	3.09 (2.62–3.64)	**2002–2019**	**−2.9** **(−3.5, −2.2)**	-	-	-	-
Age 12–20	5.55 (4.75–6.47)	5.73 (4.93–6.64)	7.16 (5.85–8.74)	6.12 (5.00–7.48)	5.12 (4.27–6.14)	4.86 (3.86–6.09)	5.87 (4.91–7.00)	5.29 (4.54–6.15)	5.36 (4.48–6.40)	4.70 (3.90–5.65)	4.10 (3.47–4.85)	3.66 (2.90–4.62)	3.24 (2.55–4.10)	3.92 (3.05–5.03)	2.47 (1.92–3.17)	2.33 (1.91–2.84)	2.51 (2.03–3.10)	2.33 (1.81–3.00)	2002–2010	−1.7 (−4.5, 1.3)	**2010–2019**	**−9.5** **(−12.3, −6.6)**	-	-
≥Age 21	4.82 (3.87–5.99)	4.69 (3.65–6.02)	4.34 (3.39–5.53)	3.87 (3.18–4.69)	4.82 (3.80–6.09)	3.86 (2.95–5.03)	4.30 (3.27–5.63)	4.73 (3.85–5.79)	4.28 (3.36–5.42)	3.42 (2.73–4.27)	3.76 (2.88–4.90)	3.79 (3.09–4.65)	3.73 (3.15–4.40)	3.74 (3.15–4.43)	3.19 (2.61–3.90)	3.78 (2.99–4.76)	3.35 (2.85–3.94)	3.28 (2.68–4.01)	**2002–2019**	**−2.0** **(−2.7, −1.2)**	-	-	-	-
NH Other	2.54 (2.02–3.19)	3.28 (2.44–4.39)	3.87 (2.82–5.28)	3.59 (2.56–5.02)	2.81 (2.14–3.68)	3.47 (2.67–4.50)	2.59 (2.10–3.19)	3.27 (2.45–4.35)	3.05 (2.37–3.91)	3.34 (2.49–4.46)	3.25 (2.65–3.98)	3.14 (2.29–4.28)	2.42 (1.87–3.12)	3.63 (2.73–4.81)	2.81 (2.25–3.51)	2.65 (2.13–3.29)	2.65 (2.19–3.21)	2.73 (2.24–3.32)	2002–2019	−0.8 (−2.0, 0.5)	-	-	-	-
Age 12–20	3.74 (2.76–5.04)	5.46 (4.11–7.22)	4.64 (3.29–6.50)	4.40 (3.34–5.78)	4.70 (3.54–6.22)	5.12 (3.64–7.16)	4.07 (3.14–5.26)	4.70 (3.65–6.03)	3.65 (2.54–5.22)	3.52 (2.56–4.84)	4.04 (3.03–5.37)	3.79 (2.81–5.09)	2.93 (2.09–4.08)	3.25 (2.45–4.31)	2.54 (1.67–3.83)	2.89 (2.09–3.98)	2.59 (1.73–3.88)	2.28 (1.60–3.24)	**2002–2019**	**−3.8** **(−5.1, −2.5)**	-	-	-	-
≥Age 21	2.29 (1.66–3.13)	2.81 (1.94–4.07)	3.70 (2.54–5.36)	3.42 (2.26–5.16)	2.42 (1.71–3.42)	3.14 (2.23–4.41)	2.30 (1.80–2.93)	2.99 (2.04–4.37)	2.93 (2.21–3.88)	3.30 (2.31–4.68)	3.09 (2.42–3.94)	3.00 (2.04–4.38)	2.32 (1.69–3.17)	3.70 (2.71–5.03)	2.87 (2.19–3.74)	2.60 (2.05–3.31)	2.66 (2.18–3.25)	2.81 (2.25–3.51)	2002–2019	0.0 (−1.3, 1.4)	-	-	-	-

^a^ Joinpoint regression is used to identify statistically significant trend change points and the APC in each trend segment using a Monte Carlo permutation method allowing for a maximum of 3 joinpoints. ^b^ APC = Annual Percentage Change; bold text indicates significance at the *p* < 0.05 level. ^c^ 95% CI = 95% confidence interval. ^d^ NH = non-Hispanic.

**Table 3 ijerph-20-06716-t003:** Number, percentage, and trend of current (past-30-day) cigar smoking, sex and age—National Survey on Drug Use and Health, United States, 2002–2019.

**Estimated Number of Cigar Smokers**																								
	**2002**	**2003**	**2004**	**2005**	**2006**	**2007**	**2008**	**2009**	**2010**	**2011**	**2012**	**2013**	**2014**	**2015**	**2016**	**2017**	**2018**	**2019**	**Joinpoint Regression Trends ^a^**					
	**Weighted N**	**Weighted N**	**Weighted N**	**Weighted N**	**Weighted N**	**Weighted N**	**Weighted N**	**Weighted N**	**Weighted N**	**Weighted N**	**Weighted N**	**Weighted N**	**Weighted N**	**Weighted N**	**Weighted N**	**Weighted N**	**Weighted N**	**Weighted N**	**Year**	**APC ^b^** **(95% CI) ^c^**	**Year**	**APC** **(95% CI)**	**Year**	**APC** **(95% CI)**
Total	12,535,430	12,744,160	13,783,780	13,623,550	13,751,740	13,264,710	13,504,580	13,255,340	13,376,990	12,773,820	13,809,180	12,441,360	12,062,450	12,392,840	12,250,060	12,528,720	12,200,270	11,818,470	-	-	-	-	-	-
Age 12–20	2,729,610	2,800,960	3,027,090	2,975,750	2,827,560	3,018,400	2,787,550	2,714,670	2,521,220	2,581,190	2,281,180	2,079,210	1,898,300	1,722,780	1,620,950	1,686,140	1,528,330	1,294,880	-	-	-	-	-	-
≥Age 21	9,805,820	9,943,200	10,756,690	10,647,810	10,924,180	10,246,310	10,717,030	10,540,670	10,855,770	10,192,630	11,528,010	10,362,160	10,164,140	10,670,060	10,629,110	10,842,580	10,671,940	10,523,590	-	-	-	-	-	-
Male	10,502,070	10,279,320	11,357,240	11,334,590	11,205,920	10,893,470	11,218,980	10,634,540	10,665,820	10,133,370	11,077,100	9,747,630	9,604,660	9,728,730	9,595,160	9,811,060	9,308,480	9,042,630	-	-	-	-	-	-
Age 12–20	2,014,560	2,095,680	2,238,820	2,232,330	2,164,020	2,269,100	2,052,650	1,988,040	1,821,310	1,768,410	1,672,270	1,545,220	1,300,250	1,155,940	1,182,970	1,162,180	1,035,300	871,840	-	-	-	-	-	-
≥Age 21	8,487,500	8,183,630	9,118,420	9,102,260	9,041,900	8,624,370	9,166,330	8,646,500	8,844,510	8,364,960	9,404,830	8,202,410	8,304,410	8,572,800	8,412,190	8,648,880	8,273,180	8,170,790	-	-	-	-	-	-
Female	2,033,370	2,464,850	2,426,550	2,288,970	2,545,820	2,371,230	2,285,610	2,620,800	2,711,170	2,650,440	2,732,090	2,693,730	2,457,790	2,664,110	2,654,900	2,717,660	2,891,790	2,775,840	-	-	-	-	-	-
Age 12–20	715,050	705,280	788,270	743,420	663,550	749,300	734,900	726,620	699,910	812,770	608,910	533,990	598,050	566,850	437,980	523,960	493,030	423,040	-	-	-	-	-	-
≥Age 21	1,318,320	1,759,560	1,638,280	1,545,550	1,882,270	1,621,940	1,550,710	1,894,170	2,011,250	1,827,670	2,123,180	2,159,740	1,859,740	2,097,260	2,216,920	2,193,700	2,398,760	2,352,800	-	-	-	-	-	-
**Prevalence of Cigar Smoking**																								
	**2002**	**2003**	**2004**	**2005**	**2006**	**2007**	**2008**	**2009**	**2010**	**2011**	**2012**	**2013**	**2014**	**2015**	**2016**	**2017**	**2018**	**2019**	**Joinpoint regression trends**					
	**% (95% Cl)**	**% (95% Cl)**	**% (95% Cl)**	**% (95% Cl)**	**% (95% Cl)**	**% (95% Cl)**	**% (95% Cl)**	**% (95% Cl)**	**% (95% Cl)**	**% (95% Cl)**	**% (95% Cl)**	**% (95% Cl)**	**% (95% Cl)**	**% (95% Cl)**	**% (95% Cl)**	**% (95% Cl)**	**% (95% Cl)**	**% (95% Cl)**	**Year**	**APC** **(95% CI)**	**Year**	**APC** **(95% CI)**	**Year**	**APC** **(95% CI)**
Total	5.33 (5.02–5.66)	5.36 (5.07–5.67)	5.73 (5.40–6.09)	5.60 (5.30–5.92)	5.59 (5.29–5.90)	5.35 (5.06–5.66)	5.41 (5.05–5.78)	5.26 (4.99–5.56)	5.27 (4.98–5.58)	4.96 (4.70–5.24)	5.31 (4.94–5.70)	4.74 (4.41–5.09)	4.55 (4.33–4.79)	4.63 (4.40–4.87)	4.55 (4.33–4.78)	4.60 (4.31–4.91)	4.46 (4.24–4.69)	4.29 (4.06–4.54)	2002–2004	4.5 (−3.6, 13.3)	**2004–2019**	**−1.9** **(−2.2, −1.6)**	-	-
Age 12–20	7.33 (6.92–7.77)	7.47 (7.10–7.87)	8.03 (7.47–8.63)	7.76 (7.36–8.18)	7.42 (7.02–7.83)	7.85 (7.34–8.38)	7.31 (6.83–7.81)	7.12 (6.64–7.64)	6.66 (6.26–7.09)	6.72 (6.27–7.19)	5.99 (5.56–6.45)	5.46 (5.07–5.87)	5.00 (4.54–5.50)	4.54 (4.10–5.03)	4.32 (3.98–4.69)	4.46 (4.07–4.89)	4.03 (3.61–4.50)	3.40 (3.12–3.70)	2002–2008	0.5 (−1.2, 2.2)	**2008–2019**	**−6.7** **(−7.5, −5.8)**	-	-
≥Age 21	4.95 (4.59–5.35)	4.97 (4.65–5.31)	5.30 (4.93–5.70)	5.20 (4.85–5.56)	5.25 (4.93–5.60 )	4.89 (4.57–5.24)	5.06 (4.67–5.49)	4.93 (4.62–5.26)	5.03 (4.70–5.38)	4.65 (4.35–4.98)	5.19 (4.78–5.64)	4.62 (4.25–5.02)	4.47 (4.23–4.74)	4.64 (4.40–4.91)	4.58 (4.32–4.86)	4.63 (4.32–4.96)	4.52 (4.28–4.78)	4.44 (4.18–4.72)	**2002–2019**	**−0.9** **(−1.2, −0.6)**	-	-	-	-
Male	9.24 (8.62–9.91)	8.94 (8.40–9.52)	9.75 (9.16–10.38)	9.61 (9.05–10.20)	9.39 (8.81–10.00)	9.06 (8.49–9.66)	9.25 (8.60–9.95)	8.70 (8.17–9.26)	8.64 (8.06–9.26)	8.13 (7.60–8.69)	8.80 (8.14–9.50)	7.67 (7.07–8.31)	7.48 (7.06–7.93)	7.50 (7.10–7.92)	7.35 (6.97–7.74)	7.43 (6.88–8.03)	7.01 (6.57–7.48)	6.77 (6.36–7.21)	2002–2005	2.1 (−2.5, 6.9)	**2005–2019**	**−2.5** **(−2.9, −2.1)**	-	-
Age 12–20	10.59 (9.86–11.36)	10.94 (10.37–11.54)	11.61 (10.73–12.56)	11.39 (10.76–12.04)	11.01 (10.28–11.78)	11.51 (10.69–12.38)	10.53 (9.80–11.30)	10.14 (9.42–10.91)	9.28 (8.67–9.93)	9.07 (8.33–9.88)	8.56 (7.92–9.25)	7.87 (7.24–8.55)	6.70 (5.94–7.55)	5.94 (5.30–6.64)	6.12 (5.48–6.82)	5.98 (5.42–6.59)	5.31 (4.70–5.98)	4.48 (3.99–5.03)	2002–2007	1.5 (−0.9, 3.9)	**2007–2019**	**−6.9** **(−7.8, −6.1)**	-	-
≥Age 21	8.97 (8.25–9.76)	8.54 (7.91–9.21)	9.38 (8.71–10.10)	9.26 (8.61–9.95)	9.07 (8.43–9.75)	8.58 (7.94–9.27)	9.01 (8.26–9.81)	8.42 (7.82–9.06)	8.52 (7.84–9.26)	7.96 (7.32–8.64)	8.84 (8.10–9.65)	7.63 (6.95–8.37)	7.62 (7.13–8.13)	7.78 (7.32–8.25)	7.56 (7.11–8.04)	7.68 (7.08–8.33)	7.30 (6.81–7.83)	7.16 (6.69–7.67)	**2002–2019**	**−1.5** **(−1.8, −1.1)**	-	-	-	-
Female	1.67 (1.51–1.85)	2.01 (1.77–2.27)	1.96 (1.75–2.18)	1.83 (1.65–2.02)	2.01 (1.80–2.24)	1.86 (1.65–2.09)	1.78 (1.58–2.01)	2.02 (1.80–2.27)	2.08 (1.82–2.39)	1.99 (1.77–2.23)	2.04 (1.84–2.26)	1.99 (1.75–2.27)	1.80 (1.61–2.00)	1.93 (1.77–2.10)	1.91 (1.74–2.10)	1.94 (1.76–2.14)	2.05 (1.87–2.25)	1.96 (1.74–2.21)	2002–2019	0.4 (−0.2, 0.9)	-	-	-	-
Age 12–20	3.93 (3.55–4.35)	3.85 (3.45–4.29)	4.28 (3.76–4.88)	3.96 (3.56–4.42)	3.59 (3.20–4.04	4.00 (3.60–4.43)	3.94 (3.47–4.47)	3.93 (3.43–4.49)	3.84 (3.37–4.38)	4.29 (3.88–4.76)	3.28 (2.82–3.80)	2.89 (2.48–3.36)	3.22 (2.77–3.74)	3.07 (2.60–3.63)	2.41 (2.01–2.89)	2.86 (2.37–3.44)	2.68 (2.21–3.25)	2.27 (1.93–2.67)	2002–2011	−0.2 (−1.7, 1.3)	**2011–2019**	**−6.4** **(−8.7, −4.1)**	-	-
≥Age 21	1.28 (1.10–1.48)	1.69 (1.43–1.99)	1.55 (1.32–1.82)	1.45 (1.26–1.67)	1.74 (1.52–1.99)	1.49 (1.27–1.74)	1.41 (1.20–1.66)	1.71 (1.48–1.96)	1.80 (1.52–2.12)	1.60 (1.37–1.88)	1.84 (1.62–2.08)	1.85 (1.58–2.16)	1.57 (1.37–1.80)	1.75 (1.58–1.94)	1.84 (1.65–2.04)	1.80 (1.62–2.00)	1.96 (1.75–2.19)	1.91 (1.67–2.19)	**2002–2019**	**1.6** **(0.9, 2.4)**	-	-	-	-

^a^ Joinpoint regression is used to identify statistically significant trend change points and the APC in each trend segment using a Monte Carlo permutation method allowing for a maximum of 3 joinpoints. ^b^ APC = Annual Percentage Change; bold text indicates significance at the *p* < 0.05 level. ^c^ 95% CI = 95% confidence interval.

**Table 4 ijerph-20-06716-t004:** Factors associated with current (past-30-day) cigar smoking, by age group—National Survey on Drug Use and Health, United States, 2020.

		Age 12–17		Age 18–20		Age 21–25		Age 26–34		Age 35+	
		aOR ^e,f^ (95% CI) ^g^	*p*-Value	aOR ^h^ (95% CI) ^f^	*p*-Value	aOR ^h^ (95% CI) ^f^	*p*-Value	aOR ^h^ (95% CI) ^f^	*p*-Value	aOR ^h^ (95% CI) ^f^	*p*-Value
**Past-30-day cigarette smoking**		**9.01** **(2.85–28.52)**	**<0.001**	**2.31** **(1.02–5.52)**	**0.045**	**5.26** **(3.43–8.06)**	**<0.001**	**3.41** **(2.22–5.22)**	**<0.001**	**4.17** **(2.80–6.19)**	**<0.001**
**Past-30-day menthol cigarette smoking ^a^**		0.27 (0.06–1.28)	0.098	1.30 (0.44–3.86)	0.631	1.27 (0.59–2.75)	0.530	1.03 (0.55–1.96)	0.917	1.41 (0.71–2.80)	0.315
**Past-30-day nicotine e-cigarette use**		1.23 (0.42–3.56)	0.703	1.36 (0.53–3.48)	0.518	**2.46** **(1.33–4.57)**	**0.005**	1.16 (0.63–2.15)	0.630	1.83 (0.90–3.69)	0.091
**Past-30-day smokeless tobacco use**		1.52 (0.16–14.63)	0.712	3.11 (0.57–16.89)	0.184	0.68 (0.35–1.35)	0.265	0.85 (0.48–1.51)	0.577	1.22 (0.54–2.76)	0.621
**Race/Ethnicity**	NH ^h^ White	1.0		1.0		1.0		1.0		1.0	
	NH Black	1.97 (0.61–6.40)	0.252	1.54 (0.58–4.08)	0.375	**3.22** **(1.97–5.27)**	**<0.001**	**2.63** **(1.45–4.75)**	**0.002**	**2.27** **(1.47–3.52)**	**<0.001**
	Hispanic	**0.17** **(0.04–0.81)**	**0.027**	**0.20** **(0.07–0.54)**	**0.002**	0.62 (0.33–1.20)	0.152	0.65 (0.35–1.20)	0.164	0.63 (0.35–1.16)	0.134
	NH Other	0.35 (0.11–1.16)	0.086	0.36(0.11–1.18)	0.089	1.53 (0.84–2.79)	0.157	0.40 (0.13–1.21)	0.102	0.50 (0.22–1.16)	0.104
**Sex**	Male	1.0		1.0		1.0		1.0		1.0	
	Female	**0.43** **(0.19–0.98)**	**0.044**	**0.45** **(0.20–1.00)**	**0.049**	**0.58** **(0.39–0.87)**	**0.009**	**0.41** **(0.25–0.67)**	**<0.001**	**0.27** **(0.19–0.40)**	**<0.001**
**Sexual identity ^b^**	Heterosexual	-	-	1.0		1.0		1.0		1.0	
	Gay or Lesbian	-	-	0.22 (0.03–1.67)	0.138	0.51 (0.13–1.94)	0.317	1.06 (0.45–2.50)	0.887	**0.18** **(0.05–0.76)**	**0.020**
	Bisexual	-	-	1.41 (0.60–3.33)	0.427	1.00 (0.60–1.69)	0.989	1.05 (0.50–2.18)	0.902	1.57 (0.74–3.32)	0.234
	Don’t know	-	-	¥	¥	1.08 (0.13–8.79)	0.943	1.44 (0.28–7.34)	0.658	1.84 (0.49–6.88)	0.357
**Completed education ^b^**	<HS diploma/GED	-	-	1.36 (0.54–3.39)	0.506	1.71 (0.94–3.11)	0.078	0.94 (0.40–2.22)	0.880	0.75 (0.37–1.54)	0.433
	≥HS diploma/GED	-	-	1.0		1.0		1.0		1.0	
**Current school grade ^c^**	≤8th grade	1.0		-	-	-	-	-	-	-	-
	9th–12th grade	**6.77** **(2.44–18.81)**	**<0.001**	-	-	-	-	-	-	-	-
**Family income**	$0–19,999	0.34 (0.03–3.79)	0.376	1.98 (0.84–4.63)	0.114	1.36 (0.78–2.37)	0.269	0.84 (0.45–1.54)	0.560	1.36 (0.78–2.38)	0.275
	$20,000–49,999	0.84 (0.26–2.68)	0.764	2.11 (0.79–5.66)	0.134	1.74 (0.93–3.25)	0.082	0.99 (0.65–1.50)	0.952	0.99 (0.60–1.65)	0.980
	$50,000–74,999	1.22 (0.32–4.59)	0.769	1.38 (0.49–3.85)	0.534	1.52 (0.80–2.89)	0.200	1.26 (0.64–2.48)	0.499	0.80 (0.45–1.41)	0.424
	≥$75,000	1.0		1.0		1.0		1.0		1.0	
**Ever military service ^b^**		-	-	¥	¥	1.19 (0.37–3.88)	0.766	0.83 (0.33–2.06)	0.676	1.45 (0.82–2.56)	0.193
**Overall health status**	Excellent/Very good/Good	1.0		1.0		1.0		1.0		1.0	
	Fair/Poor	0.95 (0.29–3.04)	0.925	0.36 (0.11–1.19)	0.093	1.19 (0.54–2.60)	0.661	1.78 (0.89–3.57)	0.101	0.89 (0.57–1.41)	0.618
**Health insurance**		0.61 (0.09–4.20)	0.609	0.83 (0.27–2.54)	0.736	0.76 (0.39–1.46)	0.397	1.52 (0.89–2.60)	0.122	1.30 (0.84–2.03)	0.235
**Past month SPD ^b,d^**		-	-	0.84 (0.40–1.77)	0.646	**1.91** **(1.20–3.02)**	**0.007**	1.10 (0.59–2.04)	0.756	**1.88** **(1.11–3.20)**	**0.020**
**Drug or alcohol use disorder**		**26.92** **(8.98–80.66)**	**<0.001**	**3.26** **(1.19–8.92)**	**0.022**	**2.58** **(1.67–4.01)**	**<0.001**	**1.70** **(1.13–2.58)**	**0.013**	**2.63** **(1.56–4.44)**	**<0.001**

Note: Bold text indicates a statistically significant finding (*p* < 0.05). ^a^ Asked among persons reporting past-30-day cigarette smoking, only. ^b^ Asked among adults ages 18 and older, only. ^c^ Asked among persons ages 12–17, only. ^d^ SPD = serious psychological distress. ^e^ aOR = adjusted odds ratio. ^f^ Adjusted for past-30-day cigarette smoking, past-30-day ENDS use, race/ethnicity, education, and drug or alcohol use disorder. ^g^ CI = confidence interval. ^h^ Adjusted for past-30-day cigarette smoking, past-30-day ENDS use, past-30-day smokeless tobacco use, sex, age, race/ethnicity, income, sexual identity, health insurance status, SPD, and drug or alcohol use disorder. ¥ = there were no observations for this cell.

## Data Availability

The data are publicly available and can be downloaded from https://www.datafiles.samhsa.gov/dataset/national-survey-drug-use-and-health-2020-nsduh-2020-ds0001 (accessed on 27 January 2023).

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
