# Peer review of "Trends and Associations of Past-30-Day Cigar Smoking in the U.S. by Age, Race/Ethnicity, and Sex, NSDUH 2002–2020"

_ijerph, 2023, doi:10.3390/ijerph20186716_

Round 1
Reviewer 1 Report
This study provides detailed analyses of cigar use in the US population based on data from the National Health Information Survey (years 2002 to 2020). Overall, the manuscript provides a concise, relevant review of the literature and a detailed descriptions of the methods, analyses, and findings. This study contributes to the literature by providing data on trends in cigar use by age, sex, and race and ethnicity. One major concern is that the analyses focused on correlates of cigar use in 2020 are not foreshadowed in the introduction and receives little attention in the discussion. As currently presented, these analyses seem tangential to the focus of this paper.
Introduction
Page 2, Line 45 – “Similarly, data from the 2022 National Youth Tobacco Survey (NYTS) showed that cigar use was the second most common tobacco product used among U.S. middle and high school students”. Suggested revision, drop “use” after cigar in this sentence as the subject seems to be the cigar product.
Page 3, line 8 – where the authors present the study questions, this reviewer recommends that the authors describe conceptually what the additional analytic methods contribute.
Methods
Page 3, line 97-99. The reader would benefit from a brief explanation of the reason that the 2020 data cannot be include in trend analysis.
The discussion is clear and provides and links the current findings to prior literature. Notably, the discussion focuses on trends in cigar use over time by age, race/ethnicity and sex, with little attention to correlates of cigar use identified in 2020.
Author Response
Reviewer 1
This study provides detailed analyses of cigar use in the US population based on data from the National Health Information Survey (years 2002 to 2020). Overall, the manuscript provides a concise, relevant review of the literature and a detailed descriptions of the methods, analyses, and findings. This study contributes to the literature by providing data on trends in cigar use by age, sex, and race and ethnicity. One major concern is that the analyses focused on correlates of cigar use in 2020 are not foreshadowed in the introduction and receives little attention in the discussion. As currently presented, these analyses seem tangential to the focus of this paper.
Response: We thank the reviewer for their kind words regarding this manuscript. While we agree that the focus of this paper was on trends in cigar smoking, changes to the 2020 NSDUH survey precluded us from including the 2020 data in the Joinpoint Regression analysis, as recommended by SAMHSA. However, given the changes to the survey in 2020, we still wanted to include it and chose to examine associations between sociodemographic correlates and cigar smoking. We edited text in the introduction to preview the analysis of 2020 NSDUH data. As noted in a response below, we also added additional language to the discussion on correlates of use.
Introduction
Page 2, Line 45 – “Similarly, data from the 2022 National Youth Tobacco Survey (NYTS) showed that cigar use was the second most common tobacco product used among U.S. middle and high school students”. Suggested revision, drop “use” after cigar in this sentence as the subject seems to be the cigar product.
Response: We have corrected this sentence to now read: “Similarly, data from the 2022 National Youth Tobacco Survey (NYTS) showed that cigars were the second most common tobacco product used among…”
Page 3, line 8 – where the authors present the study questions, this reviewer recommends that the authors describe conceptually what the additional analytic methods contribute.
Response: We have added additional information regarding what Joinpoint regression methods contribute to the analysis: “The latter method enables us to analyze trends in cigar use, using joinpoint models where several different trend lines are connected at inflection points (or “joinpoints”), rather than relying on the assumption of a linear trend line.”
Methods
Page 3, line 97-99. The reader would benefit from a brief explanation of the reason that the 2020 data cannot be include in trend analysis.
Response: We have updated the text to now read: “Due to COVID-19 pandemic-related methodological changes in data collection in 2020 (i.e., web-based data collection was implemented in Quarter 4 of 2020), these data cannot be included in trend analyses. Additional information regarding changes made to the NSDUH survey in 2020 can be found elsewhere (19). Weighted interview response rates for the NSDUH survey years were as follows: 2002 (79%), 2003 (77%), 2004 (77%), 2005 (76%), 2006 (74%), 2007 (74%), 2008 (74%), 2009 (76%), 2010 (75%), 2011 (74%), 2012 (73%), 2013 (72%), 2014 (71%), 2015 (70%), 2016 (68%), 2017 (67%), 2018 (67%), 2019 (65%), and 2020 (60%). Detailed descriptions of sampling methods and survey techniques for the 2002-2019 (20) and 2020 (19) NSDUH are available elsewhere.
The discussion is clear and provides and links the current findings to prior literature. Notably, the discussion focuses on trends in cigar use over time by age, race/ethnicity and sex, with little attention to correlates of cigar use identified in 2020
Response: We have included more detail in the paragraph of the Discussion that recaps the findings from the 2020 NSDUH analyses: “Associations between Hispanic ethnicity and cigar use were not observed among the older age groups, though the adjusted odds ratios for these groups—while not significant—were less than 1, indicating a potential similar negative association among the older age groups. This finding is consistent with those from an analysis of Wave 3 adult data of the Population Assessment of Tobacco and Health (PATH) study, which also found no association between Hispanic ethnicity—as compared to non-Hispanic White race/ethnicity—and past 30-day cigar smoking (8). These findings have the potential to inform targeted public health interventions, as well as possible legislative efforts. Ultimately, given that the 2020 survey occurred during the height of the COVID-19 pandemic—and utilized different data collection methods—these results should perhaps be interpreted with some caution. Prior research indicates that the experience of living through pandemic—whether through increased general stress or otherwise—may have impacted the prevalence and frequency of tobacco use (34, 35).”
Author Response
Reviewer 2
The paper provides a comprehensive snapshot of cigar use prevalence in the United States, demonstrating an overall decline with variations based on sociodemographic characteristics. The authors employ rigorous statistical methodology, enhancing the credibility of their findings
Abstract:
Conclusion: The abstract does not explicitly discuss the implications of the findings for public health policy or potential interventions to address the disparities observed among sociodemographic subgroups.
Response: We thank this reviewer for catching this oversight. The last sentence of the Abstract now reads: “Findings have the potential to inform cessation efforts within clinical practice, as well as regulatory efforts to reduce cigar use.”
Introduction:
While the introduction briefly mentions the significant morbidity and economic burden associated with cigar smoking (lines 1-2), it does not elaborate on the specific health risks. A brief overview of the health consequences associated with cigar smoking would help emphasize the importance of studying this topic.
Response: We have updated the referenced text to now read: “Cigar smoking is associated with significant morbidity—including oral and lung cancers, as well as heart disease (1-3)—and economic burden (4) in the United States (U.S.). It is also estimated that cigar smoking is responsible for approximately 9,000 deaths in the U.S. annually, representing 140,000 life-years lost (5).”
The introduction concludes by stating that the study builds upon prior work and extends trend analyses (lines 82-84). Still, it does not clearly state the present study's specific focus or research question. It would be helpful to provide a clear transition that explicitly states the objectives or research questions the study aims to address.
Response: We have edited the last paragraph of the Introduction to state the study’s specific research questions: “The present study assessed temporal trends in cigar smoking in the U.S. general population overall, as well as stratified by sociodemographic characteristics. … Further, the present study assessed the prevalence of cigar use in 2020 NSDUH data. The data collected in 2020 could not be included in the trend analysis given methodological changes. However, to describe U.S. cigar smokers in more recent data, this study assessed use of other tobacco products and sociodemographic characteristics of past 30-day cigar smokers.”
Methods
In lines 88-89, the author cited an external source for the detailed methodology of the NSDUH data. However, it is important to state the median survey response for each year examined.
Response: We updated the “Data source and study population” subsection to now include the weighted interview response rates for each survey year: “Weighted interview response rates for the NSDUH survey years were as follows: 2002 (79%), 2003 (77%), 2004 (77%), 2005 (76%), 2006 (74%), 2007 (74%), 2008 (74%), 2009 (76%), 2010 (75%), 2011 (74%), 2012 (73%), 2013 (72%), 2014 (71%), 2015 (70%), 2016 (68%), 2017 (67%), 2018 (67%), 2019 (65%), and 2020 (60%).”
There is a slight redundancy in the explanation of using SAS-callable SUDAAN in lines 128, 134, and 152. Although it's critical to emphasize the software and method used, perhaps consider streamlining these explanations to prevent repetitive content.
Response: We have removed the redundant references to SAS-callable SUDAAN in this subsection.
While covariates were selected based on statistical significance (p<0.05) in bivariate analysis (lines 157-158), the inclusion criteria for these could be more detailed. Also, it would be helpful to know if any consideration was given to potential multicollinearity between the covariates.
Response: We have updated the text to: “Covariates in adjusted models were identified separately for youth (age 12-17) and adults (age ≥18) and were selected based on the literature and a priori theory, as well as statistical significance (p<0.05) in bivariate analyses. Consideration was given to potential multicollinearity between covariates, investigated via examination of the correlation matrix, tolerance, and variance inflation; no evidence of multicollinearity was identified.”
Discussion
Lines 347-367: The authors highlight essential disparities in cigar smoking trends, specifically among non-Hispanic Black individuals and females aged ≥21. Further discussion on potential reasons for these increases could provide a more comprehensive picture. For example, are social, cultural, or economic factors at play that may explain these trends?
Response: Regarding these findings, we now state that “Future research can provide additional insight into the social, cultural and/or economic factors that may explain these documented disparities regarding non-Hispanic Black and female respondents aged ≥21.”
Lines 378-393: The report on 2020 data is exciting and aligns well with prior work regarding prevalence among young adults and individuals identifying as non-Hispanic Black. However, a discussion on the possible impacts of the COVID-19 pandemic on these trends in 2020 might provide more context.
Response: We have included the following text at the end of the paragraph describing cross-sectional findings from the 2020 NSDUH data: “Given that the 2020 survey occurred during the height of the COVID-19 pandemic—and utilizing different data collection methods—these results should perhaps be interpreted with some caution. Prior research indicates that the experience of living through pandemic—whether through increased general stress or otherwise—may have impacted the prevalence and frequency of tobacco use (34, 35).”
Lines 394-400: The authors could discuss why Hispanic ethnicity is associated with less current cigar use among the younger age groups but not among older age groups. Are there cultural, societal, or generational factors influencing this pattern?
Response: We have added additional discussion regarding the lack of significant associations between Hispanic ethnicity and cigar smoking among older age groups: “Associations between Hispanic ethnicity and cigar use were not observed among the older age groups, though the adjusted odds ratios for these groups—while not significant—were less than 1, indicating a potential similar trend among the older age groups. Conversely, this finding is consistent with those from an analysis of Wave 3 adult data of the Population Assessment of Tobacco and Health (PATH) study, which also found no association between Hispanic ethnicity—as compared to non-Hispanic White race/ethnicity—and past 30-day cigar smoking (8).”
Lines 401-412: The authors note several limitations of their study, including potential recall and social desirability bias, the inability to distinguish between types of cigars and the cross-sectional nature of the NSDUH survey. This is a strong start, but the discussion could be enriched by more explicitly addressing how these limitations may affect the interpretation of the results and how future research might overcome these issues. For example, would longitudinal research mitigate some of these limitations?
Response: We have included additional information in this paragraph about how some of these limitations may be overcome in the future.
While the authors have provided a sound rationale for dichotomizing the age variable to prevent small cell sizes, it might be beneficial to discuss the potential limitations of this approach in the limitations section of the paper. These could include the potential loss of information and statistical power, the somewhat arbitrary nature of choosing a cutoff, and the potential for missing non-linear relationships. Acknowledging these limitations would provide a more complete perspective on the study's methodology and results.
Response: This reviewer raises a good point regarding our decision to dichotomize the age variable in some analyses. We have updated the limitations paragraph to include possible limitations of this approach.
Overall, the manuscript provides valuable insights into cigar smoking prevalence trends over two decades. However, as stated in the abstract, the discussion lacks clear policy implications from these findings, a key component of public health research. The observed increase in cigar smoking among non-Hispanic Black individuals and females aged ≥21 especially calls for tailored interventions. The authors should consider outlining policy recommendations or identifying current policy gaps contributing to these trends. These considerations would ensure the study's findings can be effectively utilized in public health strategies and legislative decisions.
Response: We have added the following sentences within the Discussion section: “Regarding increases in cigar use prevalence among non-Hispanic Black individuals, studies have found that cigars are disproportionately marketed toward Black individuals (27, 28), and that cigars are marketed in retail outlets in neighborhoods with large proportions of Black residents (29-31). Overall, these findings point to the need for tailored interventions among those identifying as non-Hispanic Black and female persons aged ≥21. Future research can provide additional insight into the social, cultural and/or economic factors that may explain these documented disparities regarding non-Hispanic Black and female respondents aged ≥21. Continued surveillance efforts among these and other groups can contribute to tobacco control strategies, and with Food and Drug Administration (FDA) regulation, may help reduce tobacco use. In fact, in April 2022, FDA announced a proposed product standard that would prohibit characterizing flavors (other than tobacco) in cigars (32).”
Round 2
Reviewer 1 Report
The authors provided a detailed response to reviewers and have adequately addressed all reviewers' comments.